# *SPEER*: *S*entence-Level *P*lanning of Long Clinical Summaries via *E*mbedded *E*ntity *R*etrieval

**Griffin Adams**
Department of Computer Science
Columbia University
New York, NY, USA
griffin.adams@columbia.edu

**Jason Zucker & Noémie Elhadad**
Department of Biomedical Informatics
Columbia University
New York, NY
{jz2700,ne60}@cumc.columbia.edu

## Abstract

Clinician must write a lengthy summary each time a patient is discharged from the hospital. This task is time-consuming due to the sheer number of unique clinical concepts covered in the admission. Identifying and covering salient entities is vital for the summary to be clinically useful. We fine-tune open-source LLMs (Mistral-7B-Instruct and Zephyr-7B-$\beta$) on the task and find that they generate incomplete and unfaithful summaries. To increase entity coverage, we train a smaller, encoder-only model to predict salient entities, which are treated as content-plans to guide the LLM. To encourage the LLM to focus on specific mentions in the source notes, we propose SPEER: Sentence-level Planning via Embedded Entity Retrieval. Specifically, we mark each salient entity span with special "{{ }}" boundary tags and instruct the LLM to retrieve marked spans before generating each sentence. Sentence-level planning acts as a form of state tracking in that the model is explicitly recording the entities it uses. We fine-tune Mistral and Zephyr variants on a large-scale, diverse dataset of ~167k in-patient hospital admissions and evaluate on 3 datasets. SPEER shows gains in both coverage and faithfulness metrics over non-guided and guided baselines.

## 1 Introduction

Clinical professionals are experiencing physical and emotional burnout at unprecedented rates (Maslach & Leiter, 2016; National Academies of Sciences, 2019; Kroth et al., 2019). A significant factor driving clinician burnout is the Electronic Health Record (EHR): the information overload it produces, and the documentation burden it requires (Shanafelt et al., 2016; Moy et al., 2021). Increased time spent at the desk means less face-to-face interaction (only 27% of working hours spent with patients) and the resulting burnout has been associated with an increased risk of errors (Salvagioni et al., 2017; Panagioti et al., 2018).

Large Language Models (LLMs) have the potential to reduce the documentation burden by either directly replacing clinicians or acting as a co-pilot to reduce the time it takes clinicians to complete certain tasks (Clusmann et al., 2023; Perlis & Fihn, 2023). To date, much of the focus of LLMs in healthcare has been on applying closed models, such as GPT-4, to well-defined granular tasks, such as closed-book question answering (Nori et al., 2023a) and single-document radiology report summarization (Sun et al., 2023), using small publicly available benchmarks. In this paper, we adapt open-source LLMs (Zephyr and Mistral) to an onerous, longitudinal documentation task: hospital-course summarization, by training on nearly 170k patient records from a large metropolitan hospital. Hospital-course summarization (Adams et al., 2021) involves recounting in a narrative form the events occurred during the patient stay, and why they happened. Clinicians write hospital-course summaries each time a patient is discharged from a hospital in the form of the mandatory "Brief Hospital Course" section of the Discharge Summary.

Among others, the task is cognitively difficult and time consuming for two principle reasons (Adams et al., 2021). The first relates to the *identification* of salience. When a patient is admitted to a hospital, every test, diagnosis, procedure, and medication, consequential or not, is typically entered into at least one clinical note. Due to frequent copy-and-paste (Hirschtick, 2006; Adams et al., 2020), moreover, these are typically entered multiple times across notes. This leads to a severe case of note bloat (Shoolin et al., 2013) in which finding relevant information is commensurate to finding a needle in a haystack. The second major challenge involves *coverage* of salient information. Despite the high presence of redundant and irrelevant information, there is still a great deal of relevant information, especially for lengthy admissions, which can last up to a month or longer. In these cases, it is very easy to omit critical information, which could potentially render the summary clinically harmful.

Treating the notes from admission to discharge as inputs, we construct a large scale fine-tuning dataset from the full patient records for all inpatient hospital admissions at Columbia University Irving Medical Center (CUIMC) from 2020-2023. Clinician-authored Brief Hospital Course summaries are extracted from the corresponding discharge summary for the admission and serve as ground-truth references. We perform full parameter fine-tuning on Mistral and Zephyr 7B parameter models and demonstrate that they frequently hallucinate and fail to cover salient entities.

To better ground the LLMs on salient source entities, we train a specialized, smaller classification model to perform explicit content selection. Given the entity-dense nature of task, we select groups of synonymous entities—medical concepts—as the appropriate granularity for content selection. To help LLMs to adhere to these content plans, we propose **SPEER**: **S**entence-Level **P**lanning via **E**mbedded **E**ntity **R**etrieval. Specifically, we mark each salient entity span with special {{ }} boundary tags and instruct the LLM to retrieve marked spans before generating each sentence. Sentence-level planning acts as a form of state tracking in that the model explicitly records the entities it uses.

Our primary contributions are to:

- Fine-tune state-of-the-art open source LLMs (`Mistral-7B-Instruct` and `Zephyr-7B-`$\beta$) on a large-scale dataset for long-form clinical summarization, and test on three diverse datasets from different EHRs.

- Demonstrate that content selection should be thought of as its own classification task, even in the world of LLMs. Dedicated content selection, performed by a small encoder-only classifier, outperforms implicit content selection from auto-regressive LLM decoding.

- Introduce an easy-to-implement method—SPEER—which improves the coverage of salience entities and faithfulness over both non-guided and guided LLM baselines.

## 2 Related Work

**LLM Summarization.** Recent work has found that API-based closed source models, such as Claude, GPT-3, and GPT-4, can generate high quality summaries of news articles in the zero-shot setting (Goyal et al., 2022). Humans prefer LLM-generated summaries from GPT-3 over summaries generated from the previous generation of smaller, fine-tuned models (e.g., BART, PEGASUS) (Zhang et al., 2023). Human evaluation is critical to revealing the superiority of LLM-generated summaries given the limitations of reference-based metrics. In fact, Zhang et al. (2023) find that annotators judge LLM-generated summaries on par with expert-level summaries carefully crafted by freelance writers. By iteratively fusing new entities into an existing summary draft, Chain-of-Density (CoD) (Adams et al., 2023b) enables LLMs, e.g., GPT-4, to generate entity-dense summaries, which are often favored by human annotators over earlier, less dense, drafts. Explicit planning can also be performed with Summary Chain-of-Thought (SumCoT), a technique that guides LLMs to generate summaries by focusing on core news elements in a step-by-step manner, leading to more coherent and comprehensive summaries (Wang et al., 2023). On the evaluation side, Chen et al. (2023) demonstrate that zero-shot prompted LLMs can outperform existing specialized classifiers on factuality detection.

**Guided Summarization.** Abstractive summarization requires three sequential tasks: content selection (extraction), content planning (organization), and surface realization (abstraction). With simple auto-regressive generation, the first two steps are generally performed implicitly with the last. Yet, prior work suggests that making content selection an explicit step, which is handled by a separate, dedicated model, can outperform the all-in-one approach (Sharma et al., 2019). For instance, an extractive model can be used to enhance the performance of an abstractive model by treating the extract as an auxiliary input with its own encoder (GSum, (Dou et al., 2021)). Other work simply prepends salient content to the source text as a form of control (CTRLsum (He et al., 2022)). Relevant to our work, CTRLsum explores using entities as a form of control. The FROST model separates content selection and planning from realization with a single model, by having the model first generate an entity-based plan before generating the full abstractive summary (Narayan et al., 2021). **SPEER**, on the other hand, interleaves planning and realization and relies on a separately trained classifier for content selection. Adams et al. (2023a) demonstrate that offloading content selection can be used to directly control the diversity of downstream summaries. Their PGA model can be used to enhance the performance of both small encoder-decoder (BART and PEGASUS) and large decoder-only (GPT-3.5) abstractors.

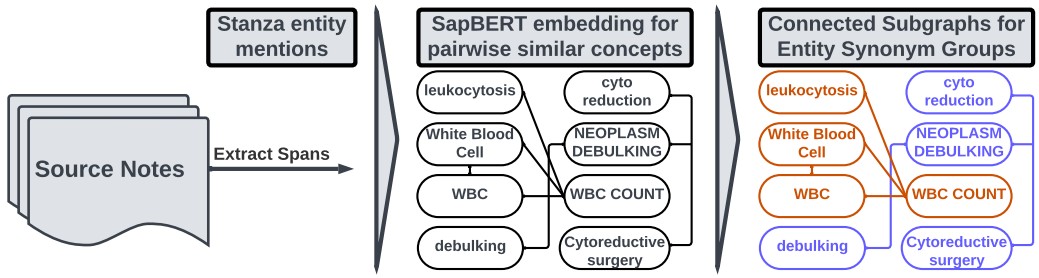

Figure 1: Extracting entities and forming groups of synonymous entities (ESGs). For each admission, we form a set of ESGs from the source notes and content selection is performed by classifying each ESG as salient or not.

**Hospital-Course Summarization.** The multi-document summarization task of synthesizing the course of events during a patient's admission to a hospital is an area of active research (Adams et al., 2021). Using a publicly available source of data (MIMIC-III), Adams et al. (2022) tackle faithfulness by re-writing, or revising, reference summaries before training models on synthetic, grounded hospital-course reference summaries. Similarly to our work, Searle et al. (2023) guide an abstractive model with clinical concepts to improve ROUGE scores of hospital-course summaries. They guide the model with all source concepts, while we perform content selection, or filtering, first. We embed the selected content back onto the source notes, whereas they guide BART with a separate encoder stream (as in GSum (Dou et al., 2021)). Preliminary studies of LLMs into biomedical applications have largely focused on well-defined medical reasoning tasks (Nori et al., 2023b;a), such as providing a differential diagnosis based on a patient record (McDuff et al., 2023). Those that have explored clinical summarization have largely focused on single-document tasks of a finer temporal granularity (Veen et al., 2023), such as generating the impressions section of a radiology report (Liu et al., 2023b;a; Chuang et al., 2023; Van Veen et al., 2023), ICD coding (Boyle et al., 2023), constructing a patient's problem list construction from a progress note (Veen et al., 2023), and generating a note from a doctor-patient conversation (Abacha et al., 2023; Ionescu et al., 2023).

## 3 Selecting Salient Entities

The process of extracting entities, identifying synonym pairs, and then forming Entity Synonym Groups (ESGs) is graphically depicted in Figure 1. In Table 1, we plot entity-based

| Statistic | Source | Reference |
|---|---|---|
| Entity Spans | 1666 | 36 |
| Unique ESGs | 473 | - |
| % Problems | 0.364 | 0.492 |
| % Treatments | 0.268 | 0.339 |
| % Tests | 0.368 | 0.169 |

Table 1: Entity and ESG statistics across source notes and reference summaries. "Entity Spans" refers to the total number of raw entity mentions, "Unique ESGs" to the number of synonym groups formed from the raw mentions across the source notes. We also report the fractional breakdown of entities by semantic type.

statistics for source notes and reference summaries. We refer to both the Figure and Table in subsequent paragraphs.

**Extracting Entities.** We use Stanza (Qi et al., 2020) for entity extraction. In particular, we use the clinical NER model (Zhang et al., 2021) which was trained on MIMIC-III notes (Johnson et al., 2016) for the i2b2-2010 clinical NER shared task (Uzuner et al., 2011). The model extracts entity spans from three disjoint semantic types: PROBLEM, TEST, TREATMENT. Problems are diagnoses and symptoms, Tests cover lab tests and imaging, while Treatments span medications and procedures.

**Identifying Entity Synonym Pairs.** Clinicians frequently rely on acronyms and shorthand when documenting, which leads to large variance in how concepts are mentioned across notes (Demner-Fushman & Elhadad, 2016; Adams et al., 2020). We follow (Adams et al., 2023d) and use similarity in embedding space to identify synonymous clusters of entity spans. Specifically, we embed all mentions with SapBERT (Liu et al., 2021)–which is trained to align synonymous clinical concepts–and use cosine similarity to identify all synonymous pairs. We manually assign binary labels (unrelated, synonymous) to 1,000 mention-pairs. We then select the threshold (0.75) for cosine similarity classification which maximizes the F1-score overlap with human labels. Figure 1 illustrates the need for semantic over lexical matching. Acronyms (WBC → White Blood Cell) have identical meanings and no lexical overlap. This also holds true for synonyms: Cytoreductive surgery and NEOPLASM DEBULKING, which both describe the resection of a tumorous growth.

**Forming Entity Synonym Groups (ESG).** For each hospital admission, we collect all entity mentions across the source notes and form a graph with one node for each unique entity mention. We assign an edge between two mentions *iff* they are exact-match duplicates or have a pairwise SapBERT similarity of $\geq 0.75$. We then treat all fully-connected sub-graphs as Entity Synonym Groups (ESG). Entity content selection is then performed over ESGs, e.g., which represent a model a single medical concept, rather than over specific span mentions. As shown in Figure 1, the process of forming ESGs by computing fully connected sub-graphs based on the pairwise similarity graph greatly reduces entity sparsity. Eight unique entity spans form two ESGs. The entities in the ESG all relate to the same precise topic–if not the exact same concept. For instance, leukocytosis is a condition characterized by a high white blood cell (WBC) count and thus is connected directly to WBC COUNT and, via the graph, to WBC and White Blood Cell.

**Defining ESG salience.** For each example (hospital admission), we extract the ESG from across the source notes. Based on the embedding similarity method for synonymity, given embedding similarity scores, we consider an ESG as "salient" if $\geq 1$ spans in the ESG is a synonym of $\geq 1$ entity span(s) extracted from the reference summary. Only 5.7% of the source ESGs are "salient" by this definition, which underscores the difficulty of content selection for this task and dataset.

**Learning ESG salience.** We build a hierarchical token-to-ESG encoder model to perform binary classification over ESGs. This approach is inspired by previous hierarchical extractive

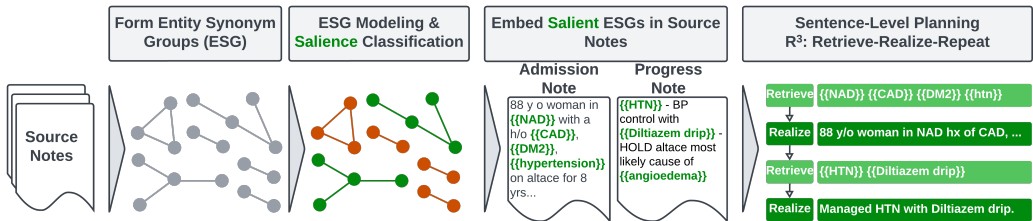

Figure 2: **SPEER**: *S*entence-Level *P*lanning via *E*mbedded *E*ntity *R*etrieval. The entire process of generating a hospital-course summary from a concatenated set of clinical notes is shown above. The first two steps relate to the formation and classification of Entity Synonym Groups (ESGs) from §1. The next two steps visually describe the SPEER approach in §4. First, salient entity mentions are marked with special {{ }} boundary tags, which indicate that they are allowed to be retrieved during generation. Then, during generation, each summary sentence is generated on its own line. Above each sentence line, the model is instructed to first retrieve the entities it plans to use in the following sentence simply by generating entities within the {{ }} tags. This single-pass decoding can be explained with the acronym *$R^3$*: **Retrieve-Realize-Repeat**, because each sentence is a realization of a plan.

methods from Liu & Lapata (2019); Bi et al. (2021); Adams et al. (2023a). First, we demarcate each entity span with newly initialized <e> and </e> tokens. Then, we concatenate all source notes and encode tokens with a long-range encoder (LongFormer (Beltagy et al., 2020)), which can fit up to 16,384 tokens. We construct hidden-state representations of each entity span by mean-pooling the hidden states of each word-piece associated with the span (inclusive of <e> and </e> tokens). Next, we mean-pool the hidden states of all entity spans associated with the same ESG (e.g., "Diabetes type 2", "DM II", "DM2"). Then, we add an ESG modeling layer as a newly initialized, fully-connected BERT encoder layer. To exploit the fact that frequently mentioned concepts tend to be salient, we assign each ESG to a numerical range according to inverse frequency (most mentions first). We learn an embedding for relative frequency and add it to the ESG hidden state before passing through the modeling layer. A linear classification head is added to produce a single logit for each modeled ESG representation. We compute a logistic loss over each ESG logit.

**ESG classification inference.** Concatenated source notes can sometimes exceed the Long-Former context window of 16, 384. During training, we simply truncate to 16, 384. Yet, during inference, to avoid information loss, we chunk notes into disjoint windows of at most 16, 384 and perform separate token-level encoding before concatenating hidden states[1]. The ESG-modeling layer has a maximum length of 1024 ESGs. For the rare case with $> 1024$ ESGs, we drop the ESGs whose term frequency of mentions across the source is lowest.

## 4   ESG-Guided Summarization

In this section, we explore methods for ESG-guided summarization: by which summary generation is conditioned on both the source documents and a pre-selected set of ESGs. We first present a simple prompt-based baseline before presenting our proposed approach, SPEER. As in (Dou et al., 2021; Adams et al., 2023a), We train models on oracle provided ESGs as guidance while performing inference with the set of model-predicted ESGs (§3).

**Prompt Guidance.** A logical approach is to convert the set of salient ESGs into a natural language prompt and instruct the model to incorporate them into its summary. To compute the oracle prompt for training, we follow the same procedure to define the labels as in §3:

---

[1]The concatenation of encoder hidden states is similar to Fusion-In-Decoder (Izacard & Grave, 2020).

"Defining ESG Salience". Then, we list out the salient ESGs by semantic type: "PROBLEMS", "TREATMENTS", and "TESTS". Each line contains each unique mention of an ESG across the source notes (delimited by ";").

**Issues with Prompt Guidance.** The above approach is simple, intuitive, and helps ground the summaries onto a set of salient entities. Yet, two issues may arise. Firstly, the model may learn to focus more on the entities themselves and not their actual usage in the source notes. The source notes are lengthy and, as such, the ratio of relevant to irrelevant content is very high. The prompt guidance, however, is precise and only includes the entities which make it into the reference summary. Relatively speaking, the model may learn to over-rely on the list of entities themselves at the expense of their usage in the source notes, which can be difficult to identify. A consequence of this would be high coverage of salient entities on the surface, yet irrelevant, inconsequential, or erroneous context supplied for these entities. Secondly, the entity guidance is extensive–some reference summaries have 100+ unique ESGs. It would be difficult even for a clinician to keep track of which ESGs have been covered so far without a discrete state tracking mechanism. State tracking is necessary for the model to both determine which of the salient ESGs have yet to be covered, and, consequently, if all ESGs are covered, to break out.

**SPEER.** To address both concerns: the lack of source note grounding and the lack of a discrete tracking mechanism, we propose **SPEER**: **S**entence-Level **P**lanning via **E**mbedded **E**ntity **R**etrieval. The **SPEER** process is shown in the last two steps of 2. To address the grounding concern, we first **E**mbed the salient **E**ntities in the source notes. To do this, we demarcate each entity span from a *salient* ESG with {{ }} boundary tags. Before generating each summary sentence, the model generates a list of the entities it plans to use, in the order in which they should appear. We refer to this **S**entence-**L**evel planning step as "**R**etrieval" because the model is performing generative retrieval over a fixed set of embedded entities. To show the model that it must only use embedded entities to form its plan, the model is taught to generate the entities with their boundary tags {{ }}. As shown in Figure 2, this can be described as $R^3$: **R**etrieve-**R**ealize-**R**epeat. $R^3$ aims to address both the grounding and tracking concerns. Firstly, by retrieving the bracketed entities, we are encouraging the model to attend to–or focus on–a specific usage of the salient ESG in context. Secondly, the act of explicitly generating the entities to include in the next sentence makes it easier to keep track of which entities have already been included in the summary. The output template for this proposed $R^3$ method of summarization is:

```
### Entities 1: {{span}} {{span}}
### Sentence 1: <sentence 1>
### Entities 2: {{span}} {{span}}
### Sentence 2: <sentence 2>
```

The final summary is simply the concatenation of each line which begins with ### Sentence. We use oracle ESGs during training and those predicted by the ESG classifier at inference. For planning, during training, we extract the in-order entity mentions from each reference sentence and add them as spans to the corresponding ### Entities line.

**Pseudo Code for SPEER.** To make it more concrete, we include pseudo code (Figure 3) for creating the training data for ESG selection (**§3**) and ESG-guided summarization, e.g. SPEER (**§4**). For ESG content selection, the model inputs are the set of input notes and the set of source ESGs. The ESG classifier is a hierarchical token-to-ESG encoder which encodes each token, before merging tokens into ESGs, performing ESG-ESG attention, and outputting a binary classification logit. For summarization, decorated inputs and reference summaries are concatenated into a single prompt and the decoder loss on the inputs is ignored.

The process for inference is similar yet relies on source ESGs *predicted* as salient by the content selector, rather than the oracle rel_src_esgs in Figure 3. The output at inference is parsed to remove planning lines, e.g., those starting with "### Entities".

Pseudo Code for generating
SPEER training data.

```
Main Script

def generate_speer_train(notes, reference):
    inputs = "\n\n".join(notes)

    # Extract Source Entities
    src_ents = EntityExtractor(inputs)

    # Cluster Source Entities into Entity Synonym Groups (ESGs)
    src_esgs = _cluster(src_ents)

    # Extract Entities from Reference Summaries
    ref_ents = EntityExtractor(reference)

    # Determine Relevant Source ESGs based on Overlap with Ref Ents
    rel_src_esgs = filter(
        lambda src_esg: _esg_relevance(src_esg, ref_ents),
        src_esgs
    )

    # Create binary classification labels based on relevance
    rel_binary = [src_esg in rel_src_esgs for src_esg in src_esgs]

    # Decorate the Inputs with the Relevant Source ESGs
    inputs_w_plan = _decorate_inputs(inputs, rel_src_esgs)

    # Decorate the Reference with the Oracle Plan (Ref Ents)
    reference_w_sent_plans = _decorate_reference(reference, ref_ents)

    # Form Training Examples for ESG selection and SPEER Summarization
    selection_ex = TrainExample(X=(inputs, src_esgs), Y=(rel_binary))
    summary_ex = TrainExample(X=inputs_w_plan, Y=reference_w_sent_plans)
    return {selection_ex, summary_ex}

data = List[Tuple(notes, reference)]
# Generate SPEER Training Datasets for Summarization & Entity Selection
summary = []
selection = []
for notes, reference in data:
    selection, summary <- generate_speer_train(notes, reference)
```

```
Helpers

def _cluster(src_ents, threshold=0.75):
    # Created an Undirected Graph with Nodes as entity spans
    graph = Graph(nodes=src_ents)

    src_embeds = Embedder(src_ents)

    for a, b in combinations(range(len(graph)), 2):
        # Compute Similarity between unique Entity Mentions
        sim = CosineSim(src_embeds[a], src_embeds[b])
        if sim >= threshold:
            graph.add_edge((a, b))

    # Entity Synonym Groups (ESGs) represent fully connected subgraphs
    esgs = graph.fully_connected_subgraphs()
    return esgs

def _esg_relevance(src_esg, ref_ents, threshold=0.75):
    src_embeds, ref_embeds = Embedder(src_esg), Embedder(ref_ents)

    # Compute Similarity between all ent spans in src_esg and ref_ents
    src_ref_sims = CosineSim(src_embeds, ref_embeds)

    # If any entity in src_esg is similar to a ref_ent, mark salient
    return src_ref_sims.max() >= threshold

def _decorate_inputs(inputs, rel_src_esgs):
    for esg in rel_src_esgs:
        for span in esg:
            inputs = inputs.replace(span, "{{" + span + "}}")
    return inputs

def _decorate_reference(reference, ref_ents):
    sents = SentenceSplitter(reference)

    # Create sentence-level plans based on entities
    sent_ents = [filter(lambda ent: ent in sent, ref_ents) for sent in sents]
    sent_plans = [
        " ".join(["{{" + ent + "}}" for ent in ents]) for ents in sent_ents
    ]

    # Interleave sentence-level plans with sentences
    return "\n\n".join([
        f"### Entities {i + 1}: {p}\n### Sentence {i + 1}: {s}"
        for p, s in zip(sent_plans, sents)
    ])
```

Figure 3: Pseudo-code to generate training data for **SPEER**, which includes a step for learning to select salient entities (**§3**), followed by entity plan-guided summarization (**§4**).

## 5   Experimental Setup

The experimental setup is detailed in Appendix §A. In summary, we train on a single dataset of ~167k admissions and test on three, diverse held out sets with ~1,000 admissions each. Two of the test sets come from the same institution as the training set, yet from different time periods and EHRs, while the third is derived from publicly available MIMIC-III notes (Johnson et al., 2016). To ensure fairness in comparisons, each model is trained for exactly the same number of steps (~14,000) in batches of 16 with the same learning rate of $5e - 6$.

**Evaluation Metrics.**   We rely on two entity-based overlap metrics: *Source-Grounded Recall* (**SGR**) for *salience* and the *Hallucination Rate* (**HR**) for *faithfulness*. To avoid penalizing models for unsupported entities in references, we separately align summary entities to entities in the sources notes, and measure overlap between source-aligned entities. Specifically, we align reference and model-generated entities to a subset of ESGs from the source notes: $\{ESG_{ref \to src}\}$ and $\{ESG_{model \to src}\}$. Then, we compute **SGR** as $SGR = \frac{|\{ESG_{ref \Rightarrow src}\} \cap \{ESG_{model \Rightarrow src}\}|}{|\{ESG_{ref \Rightarrow src}\}|}$. We compute the hallucination rate (**HR**) as $HR = \frac{|\{ENTITY_{model \nRightarrow src}\}|}{|\{ENTITY_{model}\}|}$. $|\{ENTITY_{model \nRightarrow src}\}|$ denotes the number of predicted entity mentions which do not have a corresponding source synonym. **HR** uses entity mentions and not ESGs (as in **SGR**) in order to penalize multiple hallucinations of the same synonym group. For faithfulness, we also report BERTScore-Precision (**BSP**) (Zhang et al., 2019) and **ClinDistill** (Adams et al., 2023d)—a SOTA metric for hospital-course summarization which outputs a real-value number whose mean is centered at 0. We report **ROUGE-1 / 2** despite its inverse relationship with faithfulness for clinical summarization (Adams et al., 2023c).

| Model | MIMIC | | | | | | | Average of Datasets | | | | | | |
|---|---|---|---|---|---|---|---|---|---|---|---|---|---|---|
| | Entity SGR↑ | HR↓ | BSP ↑ | Clin Distill ↑ | ROUGE R1↑ | R2↑ | # of Tokens | Entity SGR↑ | HR↓ | BSP ↑ | Clin Distill ↑ | ROUGE R1↑ | R2↑ | # of Tokens |
| **Mistral** Non-Guided | .230 | .116 | .664 | .971 | 24.3 | 6.7 | 279 | .339 | .126 | .683 | .886 | 31.9 | 16.0 | 197 |
| Guided | .236 | .171 | .648 | .541 | 23.5 | 6.2 | 352 | .401 | .151 | .678 | .683 | **33.9** | **18.1** | 251 |
| SPEER | **.302** | **.040** | **.667** | **1.240** | **25.0** | **7.0** | 324 | **.430** | **.078** | **.686** | **.947** | 33.9 | 16.6 | 234 |
| **Zephyr** Non-Guided | .245 | .121 | .653 | .899 | 25.0 | 6.8 | 335 | .386 | .138 | .673 | .789 | 33.7 | **16.4** | 257 |
| Guided | .247 | .136 | .651 | .593 | 24.0 | 6.3 | 337 | .415 | .132 | .673 | .633 | 34.0 | 16.4 | 267 |
| SPEER | **.306** | **.046** | **.662** | **1.271** | **25.9** | 7.1 | 364 | **.439** | **.084** | **.682** | **.907** | **34.1** | 16.2 | 267 |

| Model | CUIMC: 2020-2023 | | | | | | | CUIMC: 2010-2014 | | | | | | |
|---|---|---|---|---|---|---|---|---|---|---|---|---|---|---|
| | Entity SGR↑ | HR↓ | BSP ↑ | Clin Distill ↑ | ROUGE R1↑ | R2↑ | # of Tokens | Entity SGR↑ | HR↓ | BSP ↑ | Clin Distill ↑ | ROUGE R1↑ | R2↑ | # of Tokens |
| **Mistral** Non-Guided | .447 | .161 | .692 | .670 | 44.7 | 31.3 | 117 | .341 | .099 | .695 | **1.02** | 27.0 | 9.9 | 195 |
| Guided | .568 | .193 | .690 | .613 | **49.5** | **33.5** | 180 | .399 | .091 | .696 | .903 | **28.9** | **14.8** | 220 |
| SPEER | **.572** | **.117** | **.696** | **.741** | 48.4 | 32.7 | 163 | **.417** | **.075** | **.696** | .872 | 28.4 | 10.0 | 214 |
| **Zephyr** Non-Guided | .516 | .176 | .682 | .570 | 48.1 | 32.8 | 168 | .399 | .116 | .684 | **.901** | 27.9 | 9.7 | 269 |
| Guided | .582 | .152 | .684 | .554 | **49.3** | **33.1** | 203 | .417 | .107 | .685 | .758 | **28.7** | **9.8** | 260 |
| SPEER | **.588** | **.122** | **.692** | **.666** | 48.3 | 31.9 | 188 | **.424** | **.084** | **.692** | .791 | 28.2 | 9.7 | 249 |

Table 2: Results from fine-tuning `Mistral-7B-Instruct-v1` and `Zephyr-7B-`$\beta$ on **Non-Guided** and Entity-Guided (**Guided** and our proposed **SPEER**) hospital-course summarization. Metrics are defined in §5.

# 6 Results

We denote the proprietary test sets as **CUIMC:2010-2014** and **CUIMC:2020-2023**. **CUIMC** stands for Columbia University Irving Medical Center.

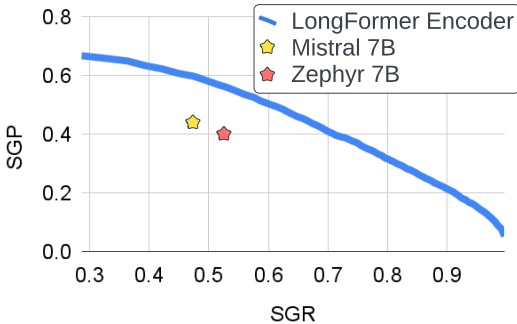

Figure 4: Comparing the entity-level performance (source-guided recall (SGR) and source-guided precision (SGP)) of *explicit* content selection (classifying entities with a LongFormer Encoder) versus *implicit* (auto-regressive decoding) on **CUIMC:2010-2014**.

**Implicit versus Explicit Content Selection.** In Figure 4, we vary the threshold for salience classification with the trained Longformer Encoder (from §3) and create a precision-recall curve, where recall is computed with SGR and precision with a similarly computed SGP(recision). On the same plot, we mark the SGP and SGR for fine-tuned **Non-Guided** Zephyr and Mistral models. Figure 4 demonstrates that Zephyr and Mistral point values fall well below the precision-recall curves of the classifier. The figure demonstrates that classification models can outperform auto-regressive models at hospital-course summary content selection, even when orders of magnitude smaller (279 million versus 7 billion parameters).

**Models which rely on entity guidance achieve higher coverage of salient entities than those that do not.** As shown in Table 2, **Guided** and **SPEER**) have higher **SGR** (source-

guided entity recall) fractions than **Non-Guided** across all dataset and base models. Looking at the average across datasets, **SGR** for models with guidance is .401/.430 for Mistral and .415/.439 for Zephyr. The **Non-Guided** model covers fewer salient entities: .339 and .386 **SGR** for Mistral and Zephyr, respectively. Summary length (**# of tokens**) plays a role for Mistral (197 < 251/234) but less of a role for Zephyr (257 < 267). For Zephyr models on **CUIMC:2010-2014**, **Non-Guided** produces the longest summaries (269 > 260/249) while also covering fewer salient ESGs: **SGR** of .399 < .417/424. While length can be controlled, the proclivity toward longer, more complete summaries may stem from the fact that guided models are provided a clear stopping criteria: to break out only when all pre-selected entities are covered.

| | Model Name | Change to Model | CUIMC: 2020-2023 | | | | | | |
| | | | Entity | | BSP | Clin ↑ | ROUGE | | # of |
| | | | SGR ↑ | HR ↓ | ↑ | Distill | R1 ↑ | R2 ↑ | Tokens |
|---|---|---|---|---|---|---|---|---|---|
| Zephyr | Non-Guided | - | .516 | .176 | .682 | .570 | 48.1 | 32.8 | 168 |
| | Guided | + Prompt Guidance | .582 | .152 | .684 | .554 | 49.3 | 33.1 | 203 |
| | Embedded | Prompt → Embedded | .574 | .147 | .688 | **.673** | 50.5 | **34.7** | 191 |
| | SPEER | + Planning with Retrieval | **.588** | **.122** | **.692** | .666 | 48.3 | 31.9 | 188 |

Table 3: From **Non-Guided** to **SPEER**: a step-by-step transition with incremental improvements in faithfulness.

**Prompt Guided is surprisingly less faithful than Non-Guided.** Across test sets and base models, prompt **Guided** summaries hallucinate more (higher **HR**) and score lower on faithfulness (**BSP**, **ClinDistill**) than **Non-Guided**. Looking at the average of datasets for Mistral, for example, **Guided** summaries have worse scores for **HR** / **BSP** / **ClinDistill** than **Non-Guided**: .151/.678/.683 versus .126/.683/.886, respectively. One would expect that instructing a model to stick to entities present in the source text would increase faithfulness. We suspect that **Guided** may learn to over-rely on the list of entities themselves at the expense of their usage in the source notes. The appended entity guidance might be stealing attention away from the source notes themselves.

**SPEER improves *both* coverage *and* faithfulness.** While adding the guidance to the prompt (**Guided**) creates a faithfulness-coverage tradeoff, **SPEER** consistently improves on both fronts. When looking at the average across datasets, the coverage of salient entities (**SGR**) is the highest for **SPEER** for both Mistral and Zephyr: .430/.439 versus .339/.386 for **Non-Guided** and .401/.415 for **Guided**. On faithfulness, **SPEER** hallucinates less: the average **HR** is .078/.084 for Mistral / Zephyr versus .126/.132 for **Non-Guided** and .151/.132 for **Guided**. Additionally, BERTScore-Precision (**BSP**) and sentence-level average faithfulness (**ClinDistill**) are highest for **SPEER**. For **BSP**, **SPEER** Mistral and Zephyr score .686/.682, more than .683/.673 for **Non-Guided** and .678/.673 for **Guided**. For **ClinDistill**, **SPEER** Mistral and Zephyr score .947/.907, more than .886/.789 for **Non-Guided** and .683/.633 for **Guided**.

**SPEER is more robust to unseen EHRs.** The model was trained on **CUIMC: 2020-2023** data, so it is unsuprising that performance is best on a held-out set of admissions from the same date range. When switching datasets and EHRs, there is a noticeable performance drop across models, especially for **MIMIC**. As discussed in Adams et al. (2022), MIMIC-III notes are highly incomplete. As such, much of the reference content is not supported by the available source notes and reference-free metrics are understandably poor. Yet, it is notable that the largest advantage (coverage and recall) for **SPEER** comes from **MIMIC**, for which the data is the noisiest and the notes come from an unseen institution. **SPEER** might be more robust to this "zero-shot" setting because it requires the least effort on the part of the abstractive component. The LLM only needs to locate {{ }} tags, rather than needing to implicitly perform salience modeling (**Non-Guided**) or to link prompted guidance back onto specific parts of the source (**Guided**).

**SPEER Ablations.** Table 3 demonstrates incremental improvements in faithfulness and coverage of salient entities as we transition from the baseline model (**Non-Guided**) to the fully loaded **SPEER** model. As discussed earlier, going from non-guided **Non-Guided**) to prompt

**Guided**) increases coverage dramatically (**SGR** goes from .516 → .582 while sentence-level faithfulness (**ClinDistill**) declines: .570 → .554. If we replace prompt guidance with embedded guidance: **Embedded**, we achieve a slight decline in **SGR**: .582 → .574 yet a decrease in hallucinations (**HR**): .152 → .147 and an increase in sentence-level faithfulness: .554 → .673. **Embedded** is **SPEER** without the sentence-level planning. The input is the same (notes with embedded salient ESGs) yet the target output is the summary without planning. Adding in planning, we arrive at **SPEER**, which leads to an increase in coverage of salient entities: .574 → .588 for **SGR** and a further decrease in hallucinations: .147 → .122.

We do note that ROUGE scores decline: ROUGE-1 from 50.5 → 48.3 but we believe that this is a necessary side effect of sentence-level planning, which encourages the model to stick to the entities in the source text and not hallucinate plausible, yet unsupported, content. Qualitatively, planning seems to cause a reduction in the number of sentences with no entities which occur in many reference summaries, yet do not contain important details. A paraphrased example is: "Patient verbalized understanding of instructions and plans to follow up with his primary doctor in two weeks." These types of sentences often achieve high ROUGE scores as they are true for many patients, but more often than not, are never stated in the source notes and cannot be assumed to be true. Including common, yet unsupported, content can artificially boost ROUGE at the expense of faithfulness.

| Model | | CUIMC: 2020–2023 Overlap w/ Guidance | | |
|---|---|---|---|---|
| | | Recall | Precision | F1 |
| | Non-Guided | .376 | .621 | .426 |
| Mistral | Guided | .596 | .695 | .621 |
| | SPEER | **.633** | **.749** | **.666** |
| | Non-Guided | .443 | .564 | .462 |
| Zephyr | Guided | .620 | .681 | .629 |
| | Embedded | .580 | .678 | .602 |
| | SPEER | **.678** | **.745** | **.691** |

Table 4: Model adherence to provided entity guidance. **Embedded** is an ablation of **SPEER** without sentence-level planning as described in the *Ablations* paragraph.

**SPEER follows the instructions better than Guided Prompt.** We compute the adherence to the instructions–which are to write a summary with a given set of ESGs–in a similar way as we measure entity based overlap between model-generated and reference entities. Specifically, we extract entities from each generated summary and align them to a subset of the source ESGs. Then, we measure the overlap (recall, precision, F1) scores vis-a-vis the guidance itself (the set of ESGs predicted as salient by the ESG classifier from §3). Table 4 demonstrates that for both Mistral and Zephyr, **SPEER** adheres better to the provided guidance. **SPEER** Mistral and Zephyr F1 score is .666/.691 versus .621/.629 for **Guided**. Even though no guidance is given, we include **Non-Guided** to illustrate how different the entities *explicitly* selected by the classifier is from the entities *implicitly* chosen during summary generation. In other terms, auto-regressive *implicit* content selection diverges from *explicit* content selection. Taken together with the results in Table 2, we believe that content selection for long-form clinical summarization is best viewed as a separate task from realization, with its own set of models, architectures, and objectives.

## 7 Conclusion

We are the first to explore fine-tuning LLMs (`Mistral-7B-Instruct` and `Zephyr-7B-`$\beta$) on the highly difficult, yet highly important, task of hospital-course summarization. We find that content selection is best performed by a dedicated salience classifier, which then guides the LLM in summary generation. We observe that simply appending the guidance to the prompt improves the coverage of salient entities yet harms faithfulness. To improve coverage *and* faithfulness, we introduce **SPEER**: *S*entence-Level *P*lanning via *E*mbedded *E*ntity *R*etrieval. By directly retrieving the entity guidance from the source notes, metrics suggest that **SPEER** summaries are more grounded and complete.

## 8 Acknowledgments

We'd like to thank Alex Fabbri and Faisal Ladhak for providing feedback on the SPEER algorithm. We'd also like to thank the anonymous COLM reviewers who carefully reviewed a draft of this paper.

This research was supported by the National Library of Medicine (NLM) and National Institute of Allergy and Infectious Diseases (NIAID) of the National Institutes of Health (NIH) under Award Number T15LM007079. The content is solely the responsibility of the authors and does not represent the official views of the NIH.

## 9 Ethics Statement

There are severe risks associated with releasing any generative AI software for deployment in a clinical setting. We do not recommend using SPEER in either fully automated or co-pilot settings without a thorough manual review of its biases and mistakes.

The CUIMC datasets contain sensitive personal health information (PHI), whose safekeeping is protected by HIPAA regulations. We adhered to best practices for safe handling of all patient materials.

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

# A  Experimental Setup

**Coarse Filtering.**   Source notes typically exceed the 8, 192 context window on which Mistral and Zephyr were trained.  Additionally, clinical notes include many irrelevant sections, including administrative text, minutely detailed descriptions of surgical procedures, and patient disclosures.  Such sections are typically easy to identify.  To filter out irrelevant content and fit the maximum context window, we learn a coarse section filter. The inputs to the extraction model are individual sections (header and body), extracted from source notes with a custom toolkit based on Clarity NLP. A RoBERTA-classifier is trained with a logistic loss to predict the salience. The labels are continuous between 0 and 1 and represent the average of ROUGE-1 and ROUGE-2 F-1 scores between the reference summary and the section text. During inference, we preserve the original order of the text but remove sections–starting with the lowest scoring–until the total tokenizer token count is no greater than 8,192.

| Dataset | Split | Example-Level Stats | | Source Stats | | Reference Stats | |
|---|---|---|---|---|---|---|---|
| | | # Admissions | Avg Length of Stay | # Notes | # Tokens | # Sentences | # Tokens |
| CUIMC:2020-2023 | Train | 167k | 6.3 days | 27.8 | 11k | 12.4 | 207.5 |
| CUIMC:2020-2023 | Test | 1k | 5.6 days | 25.5 | 13k | 11.4 | 173.9 |
| CUIMC:2010-2014 | Test | 1k | 5.2 days | 41.4 | 12k | 12.2 | 201.5 |
| MIMIC | Test | 900 | 30.8 days | 162.7 | 44k | 37.0 | 542.9 |

Table 5: Statistics for data used for training and evaluating hospital-course summarization models.  we use datasets from a large Metropolitan Hospital (**CUIMC**) at two different points of time. We also report scores on MIMIC-III, despite MIMIC having a great deal of unsupported content in reference summaries (Adams et al., 2022).

**Instruction Templates.**   As shown in Figures 6 and 7, for each model, source notes are simply concatenated chronologically. For each note, we generate a set of header lines which include the title of the note and the date of the note.  We also explicitly specify where the note lands in relation to the rest of the admission, e.g., "Day 1 of 4 (On Admission)". Although we separately fine-tune each model, we still include custom instructions.  The baseline non-guided instruction is: "Generate the BRIEF HOSPITAL COURSE summary." The prompt guidance instruction is: "Generate the BRIEF HOSPITAL COURSE summary using only the medical entities (PROBLEMS, TREATMENTS, and TESTS) provided." The guidance (list of ESGs grouped by semantic type) is appended to the source notes. SPEER's instruction is: "Retrieve a subset of the medical entities in double brackets {{  }} and use them to generate the next sentence of the BRIEF HOSPITAL COURSE summary." The line: "### BRIEF HOSPITAL COURSE:$\backslash n$" is appended to the end of the input and is an indicator to the model to start generating the summary.

**Datasets.**   We train on a single dataset and evaluate on three diverse held-out sets. **Training.** We train on ~167k in-patient hospital admissions from a large metropolitan hospital from 2020-2023. It is highly diverse in terms of patient population and care setting: emergency, surgery, obstetrics, pediatrics, etc. **Testing.** We evaluate on a held out portion of 1,000 admissions in the same time-frame as the training set: **CUIMC:2020-2023**, as well as admissions from an earlier time period: **CUIMC:2010-2014**, in which the Electronic Health Record (EHR) system was different.  As a result, note templates and titles may differ. By training on notes with one set of EHR templates and testing on both seen and unseen styles, we can test robustness of methods to subtle shifts in style and content organization. **MIMIC.** We also evaluate on a held-out set of 900 examples (source notes plus extracted reference summaries for evaluation) from publicly available MIMIC-III clinical notes with pre-processing from Adams et al. (2022). Table 5 shows high-level statistics for the train-test splits. In contrast to other commonly used clinical NLP benchmarks (Chen et al., 2022; Gao et al., 2023), the hospital-course summarization stands out as longitudinal, multi-document, and lengthy. On average, the inputs contain from 25.5 (**CUIMC**) to 162.7 (**MIMIC**) source notes, which are synthesized into lengthy (11-37 sentence) summaries. Identifying salient, non-redundant content amounts to finding many needles in many haystacks, and then

de-duplicating to ensure that each needle is unique. MIMIC admissions, on average, contain a substantially higher number of source notes than **CUIMC**, while also having very long reference summaries (37 sentences). Despite the long inputs, Adams et al. (2022) reveal much of the content in MIMIC reference summaries is not mentioned anywhere in the source notes—due to incompleteness—which reduces scores on reference-based metrics.

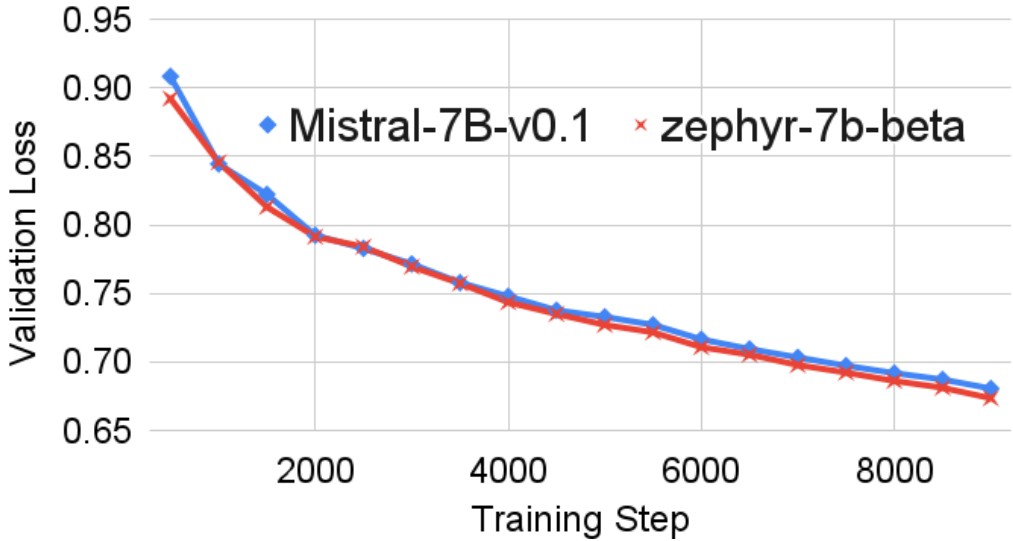

Figure 5: Validation Loss for `Mistral-7B-v0.1` and `zephyr-7b-beta` as a function of training steps across 1 epoch (covering ~167k hospital admissions).

**Training Details. ESG Content Selection.** We initialize the token-level encoder described in §3 from a 279 million parameter encoder-only model: `xlm-roberta-longformer-base-16384`, which is a Longformer initialized with weights from XLM-RoBERTa (Conneau et al., 2019) without any additional tuning. For modeling ESGs, we use a randomly initialize BERT encoder layer with the same configuration as XLM-RoBERTa. As described in §3, ESGs are sorted by inverse frequency and frequency rank embeddings are added to the representations of each ESG before passing to the modeling layer. We learn $1,024$ unique ranks and, if the source notes include more than $1,024$ ESGs, we truncate to $1,024$. We train on batches of 16 with AdamW optimizer for 100k using a scheduled learning rate (maximum $3e-5$ with linear warmup of 1000 steps, followed by linear decay). Weight decay of $5e-5$ is used. The entity boundary tags (<e>, </e>) are added as special tokens to the tokenizer and the embeddings are updated during fine-tuning. **Abstractive Summarization.** We fine-tune in two stages. The first stage involves training the baseline non-guided model for 1 epoch. For the second stage, we further-finetune the baseline and entity-guided models for an extra 4,000 steps. We break up training into two stages to reduce total training time since the weights from first stage are re-used multiple times during the second stage. *Initial Fine-Tune.* We fully fine-tune `Mistral-7B-Instruct` and `Zephyr-7B-`$\beta$ using the baseline instructions for 1 epoch with a batch size of 16 and a learning rate of $5e-6$. We used the AdamW 8-bit optimizer with a cosine learning rate scheduler. To fit the model onto two Nvidia A6000 48GB GPUs, we use DeepSpeed Stage 2 (Rasley et al., 2020), FlashAttention-2 (Dao, 2023), bfloat16 (BF16) precision, gradient checkpointing, and train on per device batches of size 1 with gradient accumulation. We used oracle section filtering to ensure that no training examples exceeded 8,192 tokens. 1 epoch took 10 days to complete. We computed a validation loss after every 500 steps of training and, in Figure 5, show a smooth loss curve. *Further Fine-Tune.* We further fine-tune all variants: baseline and entity-guided models for an additional 4,000 steps from the initial fine-tuned weights. Performance plateaus after 2500 additional fine-tuning steps. As such, we save checkpoints every 500 steps between steps 2500 and

# Mistral Instruction Templates

| Non-Guided | Guided | SPEER |
|---|---|---|

**Non-Guided**

[INST]
*Generate the BRIEF HOSPITAL COURSE summary.*

### Title: Admission Note

*DATE: 1/1/2024*
*NOTE ORDER: 1 of 2*
*DAY: 1 of 2*
*ON DAY OF ADMISSION*

**HPI:**
pt is a 90yr old w HTN

### Title: Progress Note

*DATE: 1/2/2024*
*NOTE ORDER: 2 of 2*
*DAY: 2 of 2*
*ON DAY OF DISCHARGE*

**Plan:**
pt deemed stable for discharge on ACE
[/INST]
### BRIEF HOSPITAL COURSE:

**Guided**

[INST]
*Generate the BRIEF HOSPITAL COURSE summary using only the medical entities (PROBLEMS, TREATMENTS, and TESTS) provided.*
### Title: Admission Note

*DATE: 1/1/2024*
*NOTE ORDER: 1 of 2*
*DAY: 1 of 2*
*ON DAY OF ADMISSION*

**HPI:**
pt is a 90yr old w HTN

### Title: Progress Note

*DATE: 1/2/2024*
*NOTE ORDER: 2 of 2*
*DAY: 2 of 2*
*ON DAY OF DISCHARGE*

**Plan:**
pt deemed stable for discharge on ACE

### ENTITIES
PROBLEMS:
HTN; Hypertension
TREATMENTS:
ACE; ACE inhibitors
TESTS:
[/INST]
### BRIEF HOSPITAL COURSE:

**SPEER**

[INST]
*Retrieve a subset of the medical entities in double brackets {{ }} and use them to generate the next sentence of the BRIEF HOSPITAL COURSE summary.*
### Title: Admission Note

*DATE: 1/1/2024*
*NOTE ORDER: 1 of 2*
*DAY: 1 of 2*
*ON DAY OF ADMISSION*

**HPI:**
pt is a 90yr old w {{ HTN }}

### Title: Progress Note

*DATE: 1/2/2024*
*NOTE ORDER: 2 of 2*
*DAY: 2 of 2*
*ON DAY OF DISCHARGE*

**Plan:**
pt deemed stable for discharge on {{ ACE }}
[/INST]
### BRIEF HOSPITAL COURSE:

Figure 6: Instruction Template for Mistral, which follows the syntax used during the original instruction tuning for `Mistral-7B-v0.1`.

4000 (inclusive), and report the average metric scores across this range of 4 checkpoints. We do this for robustness as there is considerable random variance for metrics across checkpoints.

**Generation Config.**   We use greedy decoding and to mitigate the problem of repetition, set a repetition penalty hyper-parameter of 1.1 (Keskar et al., 2019). We set the number of minimum tokens to be 4 and maximum new tokens to be 1,024 for the non-guided and prompted guided models. Since SPEER must generate sentence-level plans, we double the maximum new tokens from 1,024 to 2,048.

# Zephyr Instruction Templates

## Non-Guided

```
<|system|>
Generate the BRIEF
HOSPITAL COURSE
summary.
<|user|>
### Title: Admission Note

DATE: 1/1/2024
NOTE ORDER: 1 of 2
DAY: 1 of 2
ON DAY OF ADMISSION

HPI:
pt is a 90yr old w HTN

### Title: Progress Note

DATE: 1/2/2024
NOTE ORDER: 2 of 2
DAY: 2 of 2
ON DAY OF DISCHARGE

Plan:
pt deemed stable for
discharge on ACE
<|assistant|>
### BRIEF HOSPITAL
COURSE:
```

## Guided

```
<|system|>
Generate the BRIEF
HOSPITAL COURSE
summary using only the
medical entities
(PROBLEMS,
TREATMENTS, and TESTS)
provided.
<|user|>
### Title: Admission Note

DATE: 1/1/2024
NOTE ORDER: 1 of 2
DAY: 1 of 2
ON DAY OF ADMISSION

HPI:
pt is a 90yr old w HTN

### Title: Progress Note

DATE: 1/2/2024
NOTE ORDER: 2 of 2
DAY: 2 of 2
ON DAY OF DISCHARGE

Plan:
pt deemed stable for
discharge on ACE
### ENTITIES
PROBLEMS:
HTN; Hypertension
TREATMENTS:
ACE; ACE inhibitors
TESTS:
<|assistant|>
### BRIEF HOSPITAL
COURSE:
```

## SPEER

```
<|system|>
Retrieve a subset of the
medical entities in double
brackets {{ }} and use them
to generate the next
sentence of the BRIEF
HOSPITAL COURSE
summary.
<|user|>
### Title: Admission Note

DATE: 1/1/2024
NOTE ORDER: 1 of 2
DAY: 1 of 2
ON DAY OF ADMISSION

HPI:
pt is a 90yr old w {{ HTN }}

### Title: Progress Note

DATE: 1/2/2024
NOTE ORDER: 2 of 2
DAY: 2 of 2
ON DAY OF DISCHARGE

Plan:
pt deemed stable for
discharge on {{ ACE }}
<|assistant|>
### BRIEF HOSPITAL
COURSE:
```

Figure 7: Instruction Template for Zephyr, which follows the syntax used during the original instruction tuning for zephyr-7b-beta.

**Evaluation Metrics.** We rely on two entity-based overlap metrics: *Source-Grounded Recall* (**SGR**) and the *Hallucination Rate* (**HR**). Some concepts in clinical reference summaries are not present in source notes, as noted by Shing et al. (2021) and Adams et al. (2022). Unsupported–or hallucinated–reference content should not be included when computing entity overlap. As such, instead of directly computing overlap between reference and model-generated entities, we separately align summary entities to entities in the sources notes, and measure overlap between source-aligned entities. Specifically, we align reference and model-generated entities to a subset of ESGs from the source notes: $\{ESG_{ref \to src}\}$ and $\{ESG_{model \to src}\}$. Then we compute *source-grounded Recall* (**SGR**) as:

$$SGR = \frac{|\{ESG_{ref \Rightarrow src}\} \cap \{ESG_{model \Rightarrow src}\}|}{|\{ESG_{ref \Rightarrow src}\}|}$$

**SGR** does not explicitly capture entity-based faithfulness. For this, we define the hallucination rate (**HR**) as the fraction of model-generated entity mentions which do not have a source entity synonym:

$$HR = \frac{|\{ENTITY_{model \nRightarrow src}\}}{|\{ENTITY_{model}\}|}$$

where $|\{ENTITY_{model \nRightarrow src}\}|$ denotes the number of predicted entity mentions which do not have a corresponding source synonym. **HR** uses entity mentions and not ESGs (as in **SGR**) in order to penalize multiple hallucinations of the same synonym group. We report the number of tokens (**# of Tokens**) to account for length biases in the metrics.

More broadly, we capture faithfulness at both the summary-level with BERTScore-Precision (**BSP**) (Zhang et al., 2019) and, at the sentence level, with **ClinDistill** (Adams et al., 2023d)—a state of the art sentence-level faithfulness metric for hospital-course summarization. **ClinDistill** is a regression model which is distilled from an ensemble of several pre-existing state of the art faithfulness metrics. It predicts a raw, unnormalized score for each sentence, whose mean is roughly zero. We use BERTScore-Precision (**BSP**) (Zhang et al., 2019), which measures the degree to which summary tokens are well-aligned to at least one token in the source notes, rather than BERTScore-F1 because it was shown to correlate better to fine-grained expert annotations for the faithfulness of hospital course summaries (Adams et al., 2023d). Specifically, we report the **BSP** between the model-generated summary and the source notes. We compute contextualized embeddings for each token with the encoder from `allenai/led-large-16384` and follow Adams et al. (2023d) in using just the final hidden state as the representation for each token.

We also report **ROUGE-1** and **ROUGE-2** scores despite a known inverse relationship between ROUGE score and faithfulness for clinical summarization (Adams et al., 2023c). This negative correlation has to do with unsupported content in references (Adams et al., 2022). Unfaithful models will mimic unsupported content in references, while faithful models tend to stick to what is explicitly stated and, as such, be penalized by ROUGE.

