# OpenReview forum: "SPEER: Sentence-Level Planning of Long Clinical Summaries via Embedded Entity Retrieval"
_colmweb.org/COLM/2024/Conference — COLM_

### Official Review · Reviewer_3yvc · 2024-05-10

**Rating:** 7
**Confidence:** 5
**Ethics Flag:** 1

**Summary:**

This paper presents a LLM-based method, named SPEER (Sentence-level Planning via Entity Retrieval), to generate hospital-course summaries from clinical notes (admission to discharge notes). The main innovation is to identify entity synonym groups from from the notes, identify the salient entities using classification among them to perform sentence-level guiding of the LLM to generate sentences that include the salient entities. They fine-tune open-source LLMs (Mistral-7B-Instruct and Zephyr-7B) for the task. They show that SPEER improves the coverage of salient entities and faithfulness over non-guided and guided LLM baselines.

**Questions To Authors:**

- The ground truth references are extracted from discharge summaries. How noisy are these references in terms of content?
- They cluster related terms using embeddings. What would be the effect of using simple entity linking instead of identifying entity synonym groups?
-The number of entity spans in reference summaries seems too low (30; Table 1). Is this correct?

**Reasons To Accept:**

- A well-reasoned approach and a well-executed study for a difficult clinical NLP task, demonstrating good performance.
- Extensive evaluation and ablations.

**Reasons To Reject:**

- The main contribution is the sentence-level planning aspect, which may be considered to be of limited novelty to this community.
- It would have been interesting to assess the effect of the method on several biomedical summarization tasks to show generalizability.

---

> ### Author Rebuttal · Authors · 2024-05-29
>
> Thank you for the thoughtful and positive review!  We are glad that you found SPEER to be a "well-reasoned approach" showing "good performance" based on a "well-executed study for a difficult clinical NLP task".  We also appreciate your comments on our "extensive" evaluation and ablations.
>
> We believe your concerns are addressable and will address them below.
>
> ## Concerns
>
> **1. Novelty**
>
> - While planning, especially in the LLM age is a well-studied topic, our approach (SPEER) is novel along several dimensions. The first is that we perform planning by embedding-then-retrieving salient concepts into the source notes. While simple, the effect of explicitly grounding the plan in the source notes notably improves faithfulness (as shown in our Ablations).  We believe simplicity is a virtue and not a sign of limited novelty. We did not find any related literature which incorporates embedding-then-retrieval into planning.  We also would argue that forming extractive units from fully connected entity subgraphs might not be exciting to all, but is novel and, in practice, highly effective.
>
>
> **2. Other Biomedical Tasks**
>
> - We agree that adding additional tasks helps.  Given limited compute, we were unable to perform fine-tuning for more than one task.  In total, across Zephyr and Mistral, the training runs took approximately 30-40 GPU days across 2 NVIDIA RTX A6000 GPUs.  We kindly ask the reviewer to not hold this against us.  We believe that testing on 3 diverse EHR datasets demonstrates SPEER's robustness on this critical task.
>
> ## Questions
>
> **1. Noisy References**
>
> - The references are highly noisy and frequently contain content unsupported by the source notes.  Please see this paper for more quantitative analysis of the issue of noisy references for datasets extracted from real-world Electronic Health Records. Noise in references explains why we focus our findings on faithfulness and **source-grounded** recall (SGR) over ROUGE, which we admit in Section 4 is poorly suited for the task.
>
> **2. Clustering**
>
> - Clustering is necessary because ontologies are highly sparse and ontology linkers (to the UMLS) make frequent errors. The reliability of open source tools on real-world EHR datasets is a big issue for clinical NLP in general.  As such, we found that embedding based clustering with a specialized encoder substantially outperforms any method which involves brittle ontology linking. We will explain this in the camera ready and show an example!

---

> > ### Comment · Reviewer_3yvc · 2024-06-01
> > **Positive**
> >
> > The authors' responses make sense, and I keep my overall positive score.

---

### Official Review · Reviewer_ZNgw · 2024-05-11

**Rating:** 8
**Confidence:** 3
**Ethics Flag:** 1

**Summary:**

This paper introduces SPEER (Sentence-Level Planning via Embedded Entity Retrieval), a method for improving the coverage and faithfulness of long clinical summaries generated by fine-tuned large language models (LLMs). I view this work as a unique and important contribution to real-world clinical research and practice. The method and insights can potentially be generalized to other long-form summarization tasks.

**Questions To Authors:**

1. The current Hospital-Course Summarization seems to belong to a related work section.
2. A human/expert evaluation should be considered, especially regarding faithfulness analysis.

**Reasons To Accept:**

1. Fine-tuning open-source LLMs (Mistral-7B and Zephyr-7B) on a large-scale dataset for long-form clinical summarization and evaluating on three diverse test sets. This demonstrates the challenges LLMs face in generating complete and faithful summaries.
2. Showing that content selection for this task should be treated separately from content realization. A dedicated, smaller content selection classifier outperforms the implicit content selection of auto-regressive LLM decoding.
3. Proposing SPEER, a method that marks salient entities with special tags in the source text and instructs the LLM to explicitly retrieve these entities before generating each summary sentence. This sentence-level planning improves coverage of salient content and faithfulness compared to non-guided and prompt-guided LLM baselines.
4. The code and data release will be a valuable contribution to the entire community.

**Reasons To Reject:**

1. It would benefit from adding a section for related works, including state-of-the-art LLM summarization methods.
2. It would benefit from an efficiency analysis (e.g., time efficiency, scale-up analysis).

---

> ### Author Rebuttal · Authors · 2024-05-29
>
> Thank you for you thoughtful and positive review! We are glad that you find SPEER to be a **unique and important contribution to real-world clinical research and practice**.  We hope that you will help champion the paper during the discussion period.
>
> We'll address your comments below:
>
> **1. Limited related work**
>
> - We wrote a long related work section but it was *mistakenly* commented out during submission. We will add it back for camera ready. While entity-guided summarization is not new (GSum - https://arxiv.org/abs/2010.08014, FROST - https://arxiv.org/abs/2104.07606), our method is novel along several dimensions: 1) forming connected subgraphs of entities to reduce sparsity; 2) embedding the guidance in the source documents to mitigate hallucinations; 3) adding retrieval-based planning. We also discuss other more recent work on summarization with LLMs, e.g., https://arxiv.org/abs/2301.13848.
>
> **2. No efficiency analysis**
>
> - This is an important consideration, especially for real-world deployment.  Given that content to selection is performed with a very small encoder-only model, we expect it to have very minimal overhead compared to auto-regressive generation with LLMs.  We can also add scaling experiments on model size and compute (training steps).
>
> **3. No human eval**
>
> - We are finishing a large-scale human evaluation this week, which will be included in the camera ready. Preliminary results show meaningful LIKERT improvement of SPEER over the baseline: **+0.38 for completeness** and **+0.19 for faithfulness**, which agrees with our metrics. We will dedicate the extra page for camera ready to human eval and qualitative analysis of SPEER versus baseline summaries.

---

> > ### Comment · Reviewer_ZNgw · 2024-06-01
> > **Thank you for your rebuttal**
> >
> > I acknowledge that I have read the responses by the authors and I will keep my positive score.

---

### Official Review · Reviewer_HPd5 · 2024-05-20

**Rating:** 5
**Confidence:** 2
**Ethics Flag:** 1

**Summary:**

This paper presents an approach for generating clinical note summaries (in particular, discharge reports). To overcome challenges in the lengthy source text, the authors first identify important entities and then retrieve the relevance context during the generation process on the basis of these entities. Automated assessments show this process is stronger than non-guided generation using existing LLMs.

**Questions To Authors:**

- There's prior work in guiding clinical note generation with salient entities (e.g., https://www.semanticscholar.org/paper/Attend-to-Medical-Ontologies%3A-Content-Selection-for-Sotudeh-Goharian/4c3b5f8db4f44ed4d24e15227a0da30f7c20a665?utm_source=direct_link). This work seems substantially different than those due to the focus on planning, but they probably ought to be discussed as prior related work.
 - Consider using an algorithm block to describe the algorithm more clearly and precisely.

**Reasons To Accept:**

- The healthcare domain is important and an area that could benefit a lot from LLMs
- From what I can tell, the method is new and somewhat interesting. A similar approach could potentially be beneficial in other domains.
- The evaluation focused on achieving high recall and a low hallucination rate -- both of which are critical to get right in this domain
- The empirical results are strong
- The work clearly fits the remit of COLM

**Reasons To Reject:**

- The main algorithm was described imprecisely (only an English-language description and a figure). I am not certain that I would be able to reproduce the process without a clearer description.
 - The evaluation was limited to automatic assessment of the reports. It would be stronger if (at least a sample) was assessed by professionals to ensure the suitability of the reports (even if the human assessment focused on the recall and hallucination, as the automated process did)
 - The relationship between this work and prior art was not sufficiently described, especially WRT other approaches that focus on entity guidance (more below).

---

> ### Author Rebuttal · Authors · 2024-05-29
>
> Thank your for the thoughtful and positive review!
>
> We appreciate that you found the SPEER method to be “new”, “interesting”, generalizable (“beneficial in other domains”), and relevant (“clearly fits the remit of COLM”). You also agree with the focus on faithfulness and completeness, and found the empirical results to be “strong”.  Given enthusiasm for our work’s novelty, relevance, and strong results, it appears that your score (5) does not align. We responded to your concerns in an itemized list (see below), which are all easily addressable for the camera ready. We hope that your minor concerns are resolved and that you reconsider your score in light of the overall positivity of your review.
>
> **1. Reproducibility**
>
> - We agree with your suggestion to present SPEER more formally.  We will re-write the methods section such that it contains the precise notation necessary to reproduce it step-by-step.
>
> **2. No human eval**
>
> - We are finishing a large-scale human evaluation this week, which will be included in the camera ready. Preliminary results show meaningful LIKERT improvement of SPEER over the baseline: **+0.38 for completeness** and **+0.19 for faithfulness**, which agrees with our metrics. We will dedicate the extra page for camera ready to human eval and qualitative analysis of SPEER versus baseline summaries.
>
> **3. Limited related work**
>
> - We wrote a long related work section but it was *mistakenly* commented out during submission. We will add it back for camera ready. While entity-guided summarization is not new (GSum - https://arxiv.org/abs/2010.08014, FROST - https://arxiv.org/abs/2104.07606), our method is novel along several dimensions: 1) forming connected subgraphs of entities to reduce sparsity; 2) embedding the guidance in the source documents to mitigate hallucinations; 3) adding retrieval-based planning.
> - “Attend to Medical Ontologies: Content Selection for Clinical Abstractive Summarization” is a great paper and we will add it. Their work differs in that they use entities to perform ontology-aware word representations, whereas we use entities as a guide for sentence-level planning, which is more suitable for the LLM era. Similarly to GSum, Searle et al (https://arxiv.org/abs/2211.07126) guide an abstractive model with clinical concepts by adding a separate, ontology-specific encoder stream. They include all identified concepts. SPEER adds entity merging, filtering, embedding, and retrieval.

---

> > ### Author Response · Authors · 2024-06-06
> > **Follow up on the rebuttal**
> >
> > Hi - Thank you for taking the time to review SPEER.  We wanted to follow up and see if you had any lingering doubts after reading our rebuttal.  Thanks again!

---

> > > ### Comment · Reviewer_HPd5 · 2024-06-06
> > >
> > > Thanks for the detailed and reassuring response. However, given the substantial changes that you'll make to the camera-ready version of the paper, I hope you understand my hesitation in changing my assessment before reading the revision. I really look forward to the next version of the paper, though!

---

### Official Review · Reviewer_dDBd · 2024-05-23

**Rating:** 6
**Confidence:** 4
**Ethics Flag:** 1

**Summary:**

The authors address the challenge of generating comprehensive and accurate hospital discharge summaries using LLMs. Fine-tuning open-source LLMs (Mistral-7B-Instruct and Zephyr-7B-β) on nearly 170k patient records revealed issues with incomplete and unfaithful summaries. To improve performance, the authors developed SPEER (Sentence-level Planning via Embedded Entity Retrieval), which uses a smaller, encoder-only model to predict salient entities and guide the LLMs. This method marks entity spans with special tags, enhancing content coverage and faithfulness. The proposed method demonstrated improved performance over non-guided and guided baselines on three diverse datasets.

**Questions To Authors:**

1. The reviewer does not find the average total length of source documents for each example.

2. The mathematical formula for HR misses “|” on the right side.

**Reasons To Accept:**

1. The paper focuses on a critical topic in the medical domain and the proposed method is helpful in guiding future work in generating more faithful and human-quality level long-document summaries.

2. The paper clearly pointed out the challenge of selecting salient information from long clinical documents.

**Reasons To Reject:**

1. Since the final summary is the concatenation of generated summary sentences, the author provided method might affect the coherence of the final summary. However, the authors do not evaluate the coherence of the summary or mention the importance of coherence in this specific application.

2. Given that the reference summary may contain information that cannot be supported by the source document (as mentioned by the author, especially on MIMIC), it is not clear that an increase in ROUGE score by author's proposed methods mean better performance.

3. For the remaining auto-evaluation metrics, it looks like a simple baseline may also achieve a high performance. For example, suppose a model simply extracts a sentence (can extract short sentences on purpose to make the final summary short or close to the length of other model-generated summaries) that has entity mentions from each ESG based on the source document and then concatenates these extracted sentences together. This final summary would get a Source-Grounded Recall (SGR) of 1 and a Hallucination Rate (HR) of 0. Similarly, this summary would get a BertScore and ClinDistill close to 1 since all sentences are directly extracted from the source document. Would this final summary be a good one? (assuming ROUGE do not reflect the quality of LLM-generated summaries (https://arxiv.org/pdf/2209.12356, https://arxiv.org/pdf/2301.13848)).

4. The authors claim that BSP and ClinDistill are SOTA metrics for faithfulness evaluation. However, BSP do not correlate well with human judgements and it is not clear whether ClinDistill is suitable to evaluate summaries generated by LLMs as the model is initially being evaluated over non-LLM generated summaries and errors from LLM-generated summaries might be harder to detect and hence lower correlation.

5. It is not clear regarding the accuracy of forming entity synonym groups (ESG). This makes the evaluation results related to ESG not convincing enough.

6. There is a lack of significant tests in the main tables. It is not clear whether some improved results are significant.

7. There is no human evaluation of the quality of model generated summaries. Are summaries generated by the author's proposed method more coherent, readable, and factual? Do humans prefer summaries generated by author proposed methods?

8. There are no example outputs of generated summaries from different models so the reviewer cannot develop an overall feeling about the quality of those generated summaries.

---

> ### Author Rebuttal · Authors · 2024-05-29
>
> Thank you for your very thorough review!  We believe your concerns are all addressable and have itemized our responses below.
>
> **1. Coherence not evaluated**
>
> SPEER summaries are generated in-order in a single pass. We return the summary as-is without the plan tokens. We will remove the word concatenation since it falsely suggests independence. We did not evaluate coherence because ordering is relatively unimportant for the task. Despite high abstraction, summaries are written in bullet style without a meaningful order. Consequently, coherence metrics don’t work well (arxiv.org/abs/2105.00816).
>
>
> **2. ROUGE is inadequate**
>
> We include ROUGE given its historic popularity, yet downplay it in S4 for “its inverse relationship with faithfulness for clinical summarization” (arxiv.org/abs/2305.07615). We emphasize faithfulness/coverage, by which SPEER is a clear winner.
>
> **3. Simple Extraction is Enough**
>
> - Extraction does not work, unfortunately. **1) Messy Inputs.** Most notes are not written in coherent sentences, yet the summaries are. Splitting creates many one word and full note sentences. Fusion, compression, re-writing are essential. Extractive doesn't always mean  faithful (arxiv.org/abs/2209.03549). **2) Redundancy.** A salient ESG may appear 1000s of times in source: it is not enough to extract a sentence at random. Picking the right context to include requires long-context reasoning. We will add an example with an oracle extract.
>
> **4. BSP / ClinDistill Not Suitable**
>
> - BSP’s correlation to experts is similar to supervised methods on this task (Table 12 in arxiv.org/abs/2303.03948). Re ClinDistill’s generalizability, we find LLM errors are more egregious than smaller E-D models. We found these models make more non-patient specific claims (obvious extrinsic hallucinations).
>
> **5. ESG Precision**
> - We implemented ESGs because simpler methods were bad and not robust to misspellings, acronyms, etc. We will show that embedding clustering is more reliable than exact-match or ontology-based alignment (arxiv.org/abs/2208.08408).
>
> **6. Significance**
>
> - We will include CI.
>
> **7. Human evaluation**
> - We are finishing a large-scale human evaluation this week. Results show meaningful LIKERT improvement of SPEER over baseline: **+0.38 for completeness** and **+0.19 for faithfulness**, which agrees with our metrics.
>
> **8. No examples**
>
> - We cannot legally publish raw summaries. Our clinicians are working on de-identification and annotation for the camera ready.

---

> > ### Comment · Reviewer_dDBd · 2024-06-05
> >
> > Thanks for the responses. I have increased my rating from 5 to 6.

---

### Decision · Program_Chairs · 2024-07-10

**Decision:**

Accept

**Comment:**

This paper investigates the task of clinical summarization and how well fine-tuned LLMs can do on it.  They do not select the right content, so this paper augments them with a separate classification module to decide what entities to include in the summary. These entities are incorporated via a sentence-level grounding, which is a novel (to my knowledge) method of entity guidance/summary planning. Results show similar ROUGE values, but better entity coverage and reduced hallucination, which are two important metrics in this domain.

Reviewers found the problem to be worthwhile and timely, and the method interesting.  Reviewers raised concerns about aspects including the novelty, situatedness with respect to related work, evaluation, and presentation of the work. However, in the discussion, these points have largely been addressed.  The paper would benefit from related work, the human evaluation, and a more formal description of the method being added in the next version, as the authors have promised to do.